# Dreaming of the sleep lab

**Claudia Picard-Deland[1,2], Tore Nielsen[1,3]\*, Michelle Carr[4]**

**1** Dream & Nightmare Laboratory, Center for Advanced Research in Sleep Medicine, CIUSSS-NÎM – Hôpital du Sacré-Coeur de Montréal, Montréal, Canada, **2** Department of Neuroscience, Université de Montréal, Montréal, Québec, Canada, **3** Department of Psychiatry and Addictology, Université de Montréal, Montréal, Québec, Canada, **4** Department of Psychiatry, University of Rochester Medical Center, Rochester, New York, United States of America

\* tore.nielsen@umontreal.ca

**Data Availability Statement:** Source records (dream reports) cannot be shared publicly due to restrictions imposed by the Research Ethics Board of the CIUSSS du Nord-de-l'Île-de-Montréal. The data may contain identifiable information. However,

## Abstract

The phenomenon of *dreaming about the laboratory* when participating in a sleep study is common. The content of such dreams draws upon episodic memory fragments of the participant's lab experience, generally, experimenters, electrodes, the lab setting, and experimental tasks. However, as common as such dreams are, they have rarely been given a thorough quantitative or qualitative treatment. Here we assessed 528 dreams (N = 343 participants) collected in a Montreal sleep lab to 1) evaluate state and trait factors related to such dreams, and 2) investigate the phenomenology of lab incorporations using a new scoring system. Lab incorporations occurred in over a third (35.8%) of all dreams and were especially likely to occur in REM sleep (44.2%) or from morning naps (48.4%). They tended to be related to higher depression scores, but not to sex, nightmare-proneness or anxiety. Common themes associated with lab incorporation were: *Meta-dreaming*, including lucid dreams and false awakenings (40.7%), *Sensory incorporations* (27%), *Wayfinding* to, from or within the lab (24.3%), *Sleep as performance* (19.6%), *Friends/Family in the lab* (15.9%) and *Being an object of observation* (12.2%). Finally, 31.7% of the lab incorporation dreams included relative projections into a near future (e.g., the experiment having been completed), but very few projections into the past (2.6%). Results clarify sleep stage and sleep timing factors associated with dreamed lab incorporations. Phenomenological findings further reveal both the typical and unique ways in which lab memory elements are incorporated de novo into dreaming. Identified themes point to frequent social and skillful dream scenarios that entail monitoring of one's current state (in the lab) and projection of the self into dream environments elaborated around local space and time. The findings have implications for understanding fundamental dream formation mechanisms but also for appreciating both the advantages and methodological pitfalls of conducting laboratory-based dream collection.

anonymized results extracted from the dream reports will be made available upon request to the owner of the database (currently Director of the Dream & Nightmare Lab: Tore Nielsen) for researchers who meet our Ethics Review Board's (ERB) criteria for access to confidential data. Please note that our ERB does not allow external researchers to contact them directly for data requests. Requests must come via the owner of the database who holds ethical approval for the data in question. Interested researchers may also send data requests to: Tyna Paquette Research Coordinator, Center for Advanced Research in Sleep Medicine Recherche CIUSSS du Nord-de-l'Île-de-Montréal - Hôpital du Sacré-Coeur de Montréal tyna.paquette.cnmtl@ssss.gouv.qc.ca.

**Funding:** This work was supported by the Alexander Graham Bell Canada Graduate Scholarship-Doctoral Program (NSERC; https://www.nserc-crsng.gc.ca/Students-Etudiants/PG-CS/CGSD-BESCD_eng.asp; CPD), the Canadian Institutes of Health Research Grant (CIHR; MOP-115125; https://cihr-irsc.gc.ca/e/193.html; TN) and the Natural Sciences and Engineering Research Council of Canada Grant (NSERC; RGPIN-2018-05065; https://www.nserc-crsng.gc.ca/Professors-Professeurs/Grants-Subs/DGIGP-PSIGP_eng.asp; TN). This work was also supported by National Institute of General Medical Sciences grant: K12 GM106997 (MC). The funders did not play any role in the study design, data collection and analysis, decision to publish or preparation of the manuscript.

**Competing interests:** The authors have declared that no competing interests exist.

# Introduction

## The incorporation of laboratory memories in dreams

Ever since Aserinsky & Kleitman's [1] discovery of a link between vivid dreaming and rapid eye movement (REM) sleep opened the doors to laboratory investigations of dreaming, the phenomenon of *dreaming about the laboratory* has proven to be a ubiquitous feature of the research landscape (see review in [2]). These are dreams in which some element of the participant's laboratory experience reappears in the content of their subsequent dreams. The most common examples are those dreams in which episodic memory fragments of the lab reappear, e.g., the experimenter, the bedroom, polysomnography electrodes or recordings, undergoing experimental procedures, or sleeping in some other place. The dreams are typically reported when participants are intentionally awakened to report a dream in the lab, but they may also occur when participants are sleeping at home after the experiment [3, 4]. In the present study, we deal only with dreams of the lab that were dreamt and reported in the lab.

A review by Schredl [2] highlights how frequent lab references are in dreams. From a sample of twelve studies that collected dream reports from overnight REM sleep awakenings, direct laboratory references, i.e., explicit references to the experiment, were found in 6%-32% of dreams. If indirect references were also taken into account, i.e., references to sleeping, dreaming, laboratories or experiments more generally, 32%-68% of dreams incorporated some laboratory elements. This review included studies that were conducted from 1962 to 1996; the highest incorporation rates were recorded in the earlier studies and tended to decrease through the decades. According to the author, this decrease is possibly due to increasing common knowledge–and less worry–about being subjected to sleep recording techniques. It is noteworthy that since 1996, no new study with a focus on laboratory incorporation dreaming has been published.

Large inter-individual differences have also been observed in the prevalence and frequency of lab incorporations, pointing to personality factors that may be related to these dreams. For example, lab incorporations are more common among therapists than among patients or students [5–7] and are sensitive to attributes of a participant's cognitive style, such as field-dependency [8, 9]. They often depict anxious, threatening or embarrassing scenarios involving the experimenter or the experimental setting [5, 6, 8, 10] and are usually more frequent during a first night of sleeping in a lab than they are during subsequent nights [6, 11, 12].

However, as common as such dreams are, they have rarely, and not recently, been given a thorough phenomenological treatment. Nor have they been specifically assessed for the insights they might provide into how and when recent episodic memories are processed in dreams.

Here, we term a dream that contains any memory element of the sleep lab experience a *lab incorporation dream* (LID) to prevent confusion of this phenomenon with the term 'laboratory dream' which is commonly used to refer to a report of any dream that had been dreamt in the lab.

## Lab incorporations as a road to the study of memory processes in dreams

The study of LIDs may provide valuable insights into the mechanisms of dream formation. The assessment of how dream images relate to prior life experiences has been widely accepted as a tool for investigating the primary mechanisms of dream production. This 'memory source' approach to dream study has so far revealed many basic features of dreaming, such as how some memory elements are incorporated immediately and others only after a temporal delay [13, 14], or how the dreamed incorporations of memory elements are less emotionally intense than the original memories [15], or that only a small minority (0.5%–2.0%) of dreams 'replays' episodic memories in an intact episodic form [16, 17]. The memory source method has also

shown that partial or fragmentary episodic memories are very frequent in dreams. In one study, judges were confident that the dream possessed at least one memory source for over 50% of reports [16], whereas in a second study, participants themselves found dream contents to be 'obviously related' to waking life events in over 83% of reports [15]. Similarly, a review of studies of dreams bearing 'day residues' (elements of memories experienced the previous day) found that 65%-70% of reports contained such waking episodic components [18].

The contents of LIDs quite clearly draw upon elements of a participant's memories of their lab experience, reactivating, recombining and reconstituting these elements during the real-time production of the dream [16, 17, 19]. LIDs frequently portray lab-related characters, actions or settings in isolation or in combination or, as often occurs with false awakenings, in an eerily complete simulation of the lab and many of its constituents. Further study of LIDs could clearly help shed light on how memory sources are woven into novel dream scenarios and could thereby contribute to a more complete understanding of how episodic memories are processed at night (e.g., [20–22]).

## Lab experience as pre-sleep stimulation

Dream-incorporated memory sources that are studied experimentally most commonly take the form of personal, idiosyncratic life events, such as significant events recorded in home logs, or controlled pre-sleep stimulations, such as learning tasks, films or emotional stimuli. The laboratory situation neatly combines these two types of information and considering it as an experimental memory source affords several research advantages. First, participating in a sleep lab experiment is a novel and salient experience for most participants, offering an opportunity to study how dream content is influenced by a high-impact, real-life episodic event. In fact, the lab has proven to be more likely than many other stimuli to affect dream content. Specifically, lab elements appear in over a third of reports [2] whereas elements from many other types of pre-sleep stimuli, such as films, appear at rates as low as 5%-8% [23, 24]. Second, because lab procedures and paraphernalia are in many ways standardized both within a lab and from one lab to another (e.g., dream recall procedures, electrode placements, beds, monitors), results of studies from disparate protocols and research groups may be largely comparable. Moreover, such comparability could allow the combining of otherwise diverse samples into larger, more statistically powerful, data sets, as was done for the present study. Third, because the sleep lab experience is intimately familiar to experimenters who run sleep studies, elements of lab experiences are relatively easily identified and quantified as memory sources.

Here we illustrate many of these advantages in establishing a large, multi-study database of dreams collected in a single sleep lab. Combined with sleep stage scoring, we undertake both quantitative and qualitative assessments of LIDs and discuss how they can shed light on memory processes during dreaming. The specific goals of the study were to:

1. investigate state (sleep timing, sleep stages, sleep stimulation) and trait (e.g., sex, anxiety, depression, nightmare proneness) factors associated with LIDs;

2. assess the phenomenology of LIDs using a scoring system designed to capture the variety of ways laboratory elements are incorporated in dreams.

## Materials and methods

### Participants

Dream reports were drawn from the databank of the Montreal Dream & Nightmare Lab and included 403 participants (259 F, 144 M, 24.3±4.2 yrs old) from a total of 9 studies conducted

between 2006 and 2019 [3, 25–32]. All participants gave signed informed consent for laboratory polysomnography and for the recording and analysis of their dream content; all protocols were approved by the local ethics committee; secondary analysis of this database was approved as part of a larger project assessing the validity of text-mining for dream content analysis.

## Dream report collection

All participants were recorded with a standard 10–20 montage of EEG (four leads minimally: C3, C4, O1, O2; more recent projects also had F3, F4, or Fz as well) and EOG (4 electrodes at most: 2 vertical, 2 horizontal) channels referenced to A1 (re-referenced offline to A1+A2), a maximum of 4 bipolar EMG channels (chin, corrugator, arm, leg) and a maximum of 3 EKG leads. Sleep stages were scored by experts using either the American Academy of Sleep Medicine [33] or the Rechtschaffen and Kales [34] standards; in the latter case, stage 4 of sleep was rescored as stage 3 (N3) of sleep. Only the last sleep stage scored before an awakening was considered further; in most cases these awakenings were triggered by a soft tone played through loudspeakers, but were also natural awakenings in some instances. Dream reports were collected immediately following each awakening, either by recording oral reports (which were later transcribed) or by having participants directly type the reports on a keyboard in bed. A liberal definition of dream activity (anything that was going through your mind before waking up) was used in all studies.

This sample of 403 participants subjected to 650 awakenings (1 to 9 awakenings/participant; Mean = 1.6±1.2 awakenings/participant) gave reports either with a recalled dream (528 reports or 81.2%; from 343 participants; 219 F, 124 M, 24.1±4.1 yrs old), without a recalled dream (96 or 14.8%; from 90 participants), or with a white dream, i.e., a report of having had experiences during sleep prior to awakening which were subsequently forgotten (26 reports or 4.0%; from 24 participants) (see Table 1). 139 awakenings were from stage 1 (N1) sleep (104/139; 74.8% dream recall), 124 from stage 2 (N2) sleep (78/124; 62.9% dream recall), 14 from N3 sleep (9/14; 64.3% dream recall) and 373 from REM sleep (337/373; 90.3% dream recall). In order to avoid sample sizes that differ too much across sleep stages and because we had no specific hypotheses about dream incorporation as a function of NREM sleep depth, reports from N1, N2, and N3 awakenings were regrouped as NREM awakenings (N = 277).

397 awakenings were from nap studies (314/397; 79.1% dream recall from 270 participants) and 253 awakenings were from overnight studies (214/253; 84.6% dream recall from 80 participants). The precise clock time of awakening was available for 462/528 (87.5%) dream reports (missing were 68 N1, 1 N2 and 10 REM awakenings–the last sleep stage upon awakening and before dream reporting was nonetheless available for these). The time of awakenings ranged from 2:40am to 3:11pm (Mean time = 10:12am ± 151 min). In nap studies, awakenings were between 9:57am and 3:11pm (Mean time = 11:43am ± 59 min); in overnight studies they were between 2:40am and 10:10am (Mean time = 6:55am ± 114 min).

Reports were rated by participants on different dream attributes: 292 reports from 245 participants had subjective ratings for *clarity of recall*, 285 reports from 240 participants for emotional *negativity*; 284 reports from 240 participants for emotional *positivity; and* 231 reports from 193 participants for *confusion*. Each dream attribute was rated on a 1–9 scale, where 1 is not at all and 9 is a lot.

319 participants were healthy controls (272/319 reported a dream), 67 were nightmare-prone individuals, i.e., reported ≥1 nightmare or 2 bad dreams/week (55/68 reported a dream), and 17 were experienced Vipassana meditators (16/17 reported a dream). Depression (Beck Depression Index [35]) and anxiety (State Trait Anxiety Inventory–STAI [36]) scores were available for 253 participants who recalled a dream.

**Table 1. Frequencies of dream recall and LIDs by sleep stage and sleep timing.**

| | Total | | REM | | NREM | | Overnight | | Nap | |
|---|---|---|---|---|---|---|---|---|---|---|
| | # | % | # | % | # | % | # | % | # | % |
| #Awakenings | 650 | | 373 | | 277 | | 253 | | 397 | |
| No recall | 96 | 14.8 | 29 | 7.8 | 67 | 24.2 | 27 | 10.7 | 69 | 17.4 |
| White Dreams | 26 | 4.0 | 7 | 1.9 | 19 | 6.9 | 12 | 4.7 | 14 | 3.5 |
| Dream recall | 528 | 81.2 | 337 | 90.3 | 191 | 69.0 | 214 | 84.6 | 314 | 79.1 |
| LIDs[a] | 189 | 35.8 | 149 | 44.2 | 40 | 20.9 | 37 | 17.3 | 152 | 48.4 |
| **LID elements[b]** | | | | | | | | | | |
| People | 103 | 54.5 | 80 | 53.7 | 23 | 57.5 | 11 | 29.7 | 92 | 60.5 |
| Place | 142 | 75.1 | 115 | 77.2 | 27 | 67.5 | 20 | 54.1 | 122 | 80.3 |
| Objects | 72 | 38.1 | 55 | 36.9 | 17 | 42.5 | 10 | 27.0 | 62 | 40.8 |
| Task | 102 | 54 | 83 | 55.7 | 19 | 47.5 | 13 | 35.1 | 89 | 58.6 |
| Sleep-related activities | 17 | 9 | 17 | 11.4 | 0 | 0.0 | 9 | 24.3 | 8 | 5.3 |
| **LID themes[b]** | | | | | | | | | | |
| Sleep Performance | 37 | 19.6 | 29 | 19.5 | 8 | 20.0 | 6 | 16.2 | 31 | 20.4 |
| Wayfinding | 46 | 24.3 | 38 | 25.5 | 8 | 20.0 | 2 | 5.4 | 44 | 28.9 |
| Meta-dreaming | 77 | 40.7 | 64 | 43.0 | 13 | 32.5 | 5 | 13.5 | 72 | 47.4 |
| False awakening/waking | 69 | 36.5 | 57 | 38.3 | 12 | 30.0 | 5 | 13.5 | 64 | 42.1 |
| Friends/Family in lab | 30 | 15.9 | 25 | 16.8 | 5 | 12.5 | 3 | 8.1 | 27 | 17.8 |
| Sensory | 51 | 27 | 41 | 27.5 | 10 | 25.0 | 2 | 5.4 | 49 | 32.2 |
| Object of observation | 23 | 12.2 | 17 | 11.4 | 6 | 15.0 | 5 | 13.5 | 18 | 11.8 |
| Temporal orientation All (IF+NF+P) | 63 | 33.3 | 53 | 35.6 | 10 | 25.0 | 6 | 16.2 | 57 | 37.5 |
| All Future (IF+NF) | 60 | 31.7 | 51 | 34.2 | 9 | 22.5 | 6 | 16.2 | 54 | 35.5 |
| Immediate Future (IF) | 50 | 26.5 | 42 | 28.2 | 8 | 20.0 | 3 | 8.1 | 47 | 30.9 |
| Near Future (NF) | 19 | 10.1 | 18 | 12.1 | 1 | 2.5 | 3 | 8.1 | 16 | 10.5 |
| Past (P) | 5 | 2.6 | 4 | 2.7 | 1 | 2.5 | 1 | 2.7 | 4 | 2.6 |

LIDs: lab incorporation dreams; #: Total count; %: Percentages. IF: Immediate future; NF: Near future; P: Past.

[a] % calculated on the total number of dream recall;

[b] % calculated on the total number of LIDs.

Four of the 9 studies included some manipulation during sleep: 60 reports from 19 participants were from overnight sleep with partial REM sleep deprivation; 15 reports from 15 participants were from naps during which a pressure cuff was applied on one leg for no more than 15 minutes; 55 reports from 55 participants were from naps during which 40 Hz tACS was applied to frontal regions in REM sleep for ~2.5 minutes; 46 reports from 46 participants were from naps during which auditory stimulation (targeted memory reactivation) was applied for ~5 minutes in NREM sleep (n = 20) or REM sleep (n = 26). The ensemble of these 176 dream reports was grouped under the binary variable Sleep manipulation (present vs. absent).

## Qualitative assessment

The different steps that led to scoring of the dream reports are summarized in '*Scoring of Laboratory Incorporation Dreams (SoLID) Criteria*' (S1 Appendix).

First, 2 judges working independently (J1: ASR; J2: CPD) scored each dream report for whether (1) or not (0) it incorporated any memory element of the laboratory as described in categories 1–5 of the SoLID system (people, places, objects, tasks, sleep-related activities).

Second, 3 judges (J2: CPD, J3: TN, J4: MC) separately scored each LID (i.e., dream scored as 1 in the previous step) for the presence (1) or absence (0) of each of the same 5 lab memory elements. Four dreams originally identified as incorporating the lab (out of 193) were rescored as 0 since the 3 judges agreed that they did not clearly incorporate any elements from the 5 categories.

Third, 3 judges (J2, J3, J4) together identified and defined 6 themes that were frequently observed in LIDs (categories 6–11 of the SoLID system). They then separately scored each LID for the presence (1) or absence (0) of each of the 6 themes. They also scored each dream on a global incorporation Likert scale of 1–7, rating the overall impression that the dream was influenced by the laboratory (LID richness). Finally, they scored each LID for Temporal orientation (categories 12–14 of the SoLID system).

Reliability estimates between each pair of judges were then calculated. Kappa estimates ranged between 0.189–0.873 (Mean = 0.646) for J2 and J3; between 0.364–0.843 (Mean = 0.577) for J2 and J4, and between 0.289–0.824 (Mean = 0.497) for J3 and J4. Scoring reliabilities between J2 and J3 were overall strongest. Therefore, categories for which kappa estimates were deemed moderate to high between J2 and J3 (People: 0.873; Places: 0.786; Objects: 0.651; Task: 0.639; Sleep Performance: 0.860; Wayfinding: 0.647; Friend/Family in lab: 0.943; Meta-dreaming: 0.683; Global Score: 0.858) were retained for further analysis using J2's scores as dependent measures. For categories with low reliability estimates (Sleep activities: 0.468; Sensory: 0.454; Temporal orientation: 0.189, Observer Effect: 0.337), the 3 judges together refined the definition of those categories and worked through each dream report to a consensus; the consensus scores were used as dependent measures.

## Statistical analyses

Statistical analyses were conducted in R. Results visualizations were performed with R and GraphPad Prism 9.

Generalized linear mixed models (GLMM; *glmer* function [37]) were used to assess factors that predict the occurrence of LIDs (1 or 0). We entered Sleep stage (REM vs NREM), Sleep timing (overnight vs nap), Sleep manipulation (present vs absent), and Sex (M vs F) as fixed effects predictors. As exploratory analyses, we also added factors that were available in a smaller number of participants (N = 253): Nightmare proneness (control vs nightmare-prone), Trait anxiety (STAI-Trait), State anxiety (STAI-State) and Depression (BDI) scores as fixed effects in the model. To further disentangle the contribution of different factors, we used Sleep stage, Sleep timing and Clock time of awakening as factors in independent models.

GLMM were also used to assess factors that predicted occurrences of the 6 different LID themes (1 or 0) separately, with Sleep stage, Sleep timing and Nightmare proneness as fixed effects. Finally, one GLMM was used to assess the occurrence of Future temporal orientation (1 or 0), with Clock time of awakening and Sleep stage as fixed effects.

Linear mixed models (LMM; *lmer* function) were used to assess continuous outcomes (*global incorporation score*; *negativity*; *positivity*; *confusion* and *clarity of recall*) with LIDs, Sleep stage, Sleep timing and Nightmare proneness as fixed factors.

Participant# was added as a random effect in all GLMM and LMM models to take into account variable numbers of dreams reported by individual participants. GLMM were fit by maximum likelihood (Laplace Approximation) and LMM were fit by restricted maximum likelihood.

## Results

### Lab incorporation dreams are common and pervasive

LIDs occurred in over a third of all dreams (189/528; 35.8%) and were reported by almost half of all participants who recalled a dream (164/343; 47.8%) (Table 1). They occurred in all stages of sleep: N1 (17/104; 16.4%), N2 (21/78; 26.9%), N3 (2/9; 22.2%), NREM combined (40/191; 20.9%), REM (149/337; 44.2%).

### Lab incorporation dreams are more probable from morning naps and REM sleep

GLMM assessing factors predicting the occurrence of LIDs were conducted on 528 reports from 343 participants.

Sleep stage (β = -1.142, SD = 0.259, z = -4.416, p < .0001) and Sleep timing (β = -1.561, SD = 0.261, z = -5.987, p < .00001) both predicted the presence of LIDs, while Sleep manipulation (β = -0.130, SD = 0.233, z = -0.556, p = 0.578) and Sex (β = -0.2250, SD = 0.232, z = -0.969, p = 0.332) did not.

Specifically, LIDs were more frequent in REM (149/337; 44.2%) than NREM (40/191; 20.9%) dreams, and more frequent in naps (152/314; 48.4%) than overnight awakenings (37/214; 17.3%) (Fig 1B and Table 1). They were not related to Sleep manipulation (present: 40.3%; absent: 33.5%) or Sex (F: 37.9% of reports, 42.8% of participants; M: 31.9% of reports, 38.2% of participants).

Because REM sleep is more prominent in the morning than in preceding sleep cycles, we further assessed whether the nap effect on LIDs was associated with the timing of REM sleep. An interaction effect could not be directly computed because of convergence problem introduced by the complexity of the model. However, when using Sleep stage as the only predictor, REM dreams were still more often LIDs than were NREM dreams in both nap (REM: 119/215; 55.3%; NREM: 33/99; 33.3%; β = -1.020, SD = 0.303, z = -3.368, p = .0008) and overnight (REM: 30/122; 24.6%; NREM: 7/92; 7.6%; β = -1.440, SD = 0.479, z = -3.004, p = .003) studies separately. Similarly, when using Sleep timing as the only predictor, nap dreams were still more often LIDs than were overnight dreams for both REM (β = -1.420, SD = 0.294, z = -4.831, p < .0001) and NREM (β = -1.804, SD = 0.447, z = -4.032, p < .0001) sleep. These

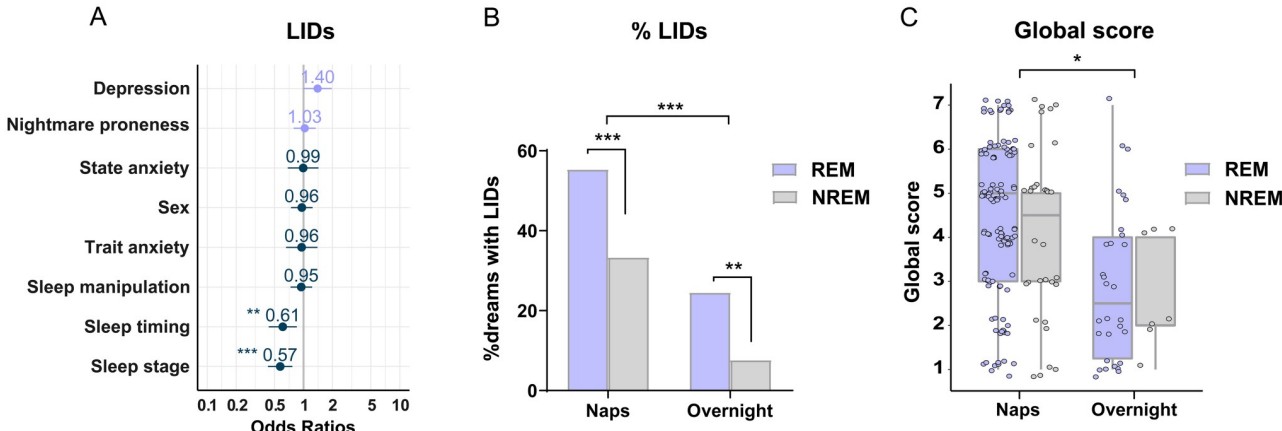

**Fig 1. Factors predicting the occurrence of lab incorporation dreams (LIDs).** A) Sleep timing (naps > overnight) and Sleep stage (REM > NREM) were the only predictors of LIDs. B) Independent main effects of Sleep timing and Sleep stage on percentage of LIDs. C) Global incorporation score for LIDs as a function of Sleep timing and Sleep stage. Error bars represent 95% confidence intervals. **p < .01, ***p < .001.

results suggest that stage and timing predictors influence LIDs independently (Fig 1B), even though an interaction between these factors is not fully excluded from the results. Clock time of awakening alone was also predictive of LIDs (p < .001), with later awakenings being more likely to have LIDs. However, when adding Sleep Stage and Sleep timing as predictors, only Sleep stage (p = .0009) and Sleep timing (p = .0001), but not Clock time of awakening (p = .102), predicted the presence of LIDs.

To explore other participant factors, we added Nightmare proneness (control vs. nightmare-prone participants), State anxiety, Trait anxiety and Depression scores (available in a smaller sample of participants; $N_{subj}$ = 253; $N_{obs}$ = 345) as predictors in the model: higher depression scores were marginally related to the presence of LIDs (β = 0.048, SE = 0.025, z = 1.932, p = 0.053), while anxiety scores (both p>.825) and Nightmare proneness (p = .840) were not. Sleep stage (p < .001) and Sleep timing (p = .004) continued to independently predict LIDs, while Sleep manipulation (p = .693) and Sex (p = .753) did not (Fig 1A).

## Lab incorporations are richer in dreams from later awakenings

We used LMM to assess which factors predicted LID global incorporation scores, with Sleep stage, Sleep timing, Nightmare proneness, Sleep manipulation and Sex as predictors. Only Sleep timing predicted higher LID global scores, with nap dreams (4.38±1.8 out of 7) having a greater mean global score than overnight (2.89±1.66) dreams (β = -1.252, SD = 0.362, $t$(137.8) = -3.460, p = .0001) (Fig 1C). All other predictors were non-significant (all p>.169). Clock time of awakening alone also predicted LID global scores, with higher global scores being attributed to dreams recalled later in the morning (β = 0.285, SD = 0.072, $t$(149.6) = 3.927, p = .0001)–although this relationship did not hold when looking separately for overnight (p = .916) and nap (p = .274) awakenings.

In sum, LIDs were strongly associated with some state factors, being more likely in dreams from REM sleep and from morning naps; they were also marginally more likely with higher trait depression, but were not related to trait or state anxiety, sex, or nightmare proneness. Moreover, later awakenings predicted both the occurrence and the richness of LIDs, but this effect was confounded by the type of sleep protocol used (overnight vs nap). However, as the clock time of awakenings did not vary much within overnight or nap studies, a circadian influence on this phenomenon cannot be ruled out.

## General attributes of lab incorporation dreams

We used LMM to assess whether presence of a LID was a significant predictor of 4 different dream attributes (negativity, positivity, confusion, clarity of recall). As predictors, we entered LID (present or absent), Sleep stage, Sleep timing and Nightmare proneness. Although most results would not survive a strict error correction threshold (p < .01), several effects were at least suggested for each of the attributes. Higher negativity was predicted by LID presence (β = 0.682, SE = 0.291, $t$(276.5) = 2.346, p = .019), but not by Sleep stage (p = 0.527), Sleep timing (p = 0.615) or Nightmare proneness (p = 0.244). Higher positivity was marginally predicted by Sleep stage (REM>NREM; β = -0.514, SE = 0.302, $t$(278.9) = -1.701, p = .090), but not by LID (p = .122), Sleep timing (p = 0.466) or Nightmare proneness (p = 0.680). Higher confusion was predicted by LID presence (β = 1.143, SE = 0.323, $t$(217.6) = 3.534, p < .001), but not by Sleep stage (p = .142), Sleep timing (p = 0.305) or Nightmare proneness (p = 0.138). Higher clarity of recall was predicted by LID presence (β = 1.270, SE = 0.254, $t$(287.0) = 5.000, p < .0001), Sleep stage (REM>NREM; β = -1.552, SE = 0.278, $t$(287) = -5.579, p < .0001) and marginally by Nightmare proneness (β = -0.551, SE = 0.290, $t$(287.0) = -1.902, p = .058), but not by Sleep timing (p = 0.936).

Because clarity of recall was higher for LIDs and could therefore influence the salience of other dream attributes, we added it as an additional fixed factor to the models predicting negativity, positivity, and confusion. Both positivity and confusion were predicted by LID absence and presence, respectively (all p < .01), as well as by clarity of recall (all p < .01); whereas negativity was predicted by clarity of recall (p = .003), but not by LID presence (p = .162).

In sum, compared with non-LIDs, LIDs were more clearly recalled yet also more subjectively confusing. They were also less emotionally positive when controlling for their recall clarity. While they tended to be more negative, this effect was confounded by their greater clarity of recall.

## Qualitative assessment of lab incorporation dreams

**Common themes.** A qualitative assessment of LIDs revealed six common themes, including *Meta-dreaming* (40.7% of LIDs), *Sensory incorporations* (27%), *Wayfinding* to, from or within the lab (24.3%), *Sleep as performance* (19.6%), *Friends/Family in the lab* (15.9%) and *Object of observation* (12.2; see Table 1 for frequencies and Table 2 for examples). 53 LIDs (28%) did not correspond to any of these themes, 52 LIDs (27.5%) had only 1 theme, 50 (26.5%) had 2 themes, 25 (13.2%) had 3 themes, 8 (4.2%) had 4 themes, 1 (0.5%) had 5 themes and none had all 6 themes (see Table 3 and S2 Appendix for examples of dreams with multiple themes).

The six themes are defined in more detail in S1 Appendix and examples are provided in Table 2. *Meta-dreaming* includes dream experiences that involve an alteration in consciousness or bodily ownership such as false awakenings, pre-lucid or lucid dreams, sleep paralysis, out-of-body experiences, dreaming within a dream, and similar alterations. *Sensory incorporations* refer to real bodily sensations that occur during sleep and make their way into dreams, such as the feeling of electrodes on the skin, muscle atonia or weakness, being cold, thirsty or hungry, having to urinate or hearing sounds coming from speakers in the bedroom. *Wayfinding* refers to the dreamer ambulating in, around, into or out of the lab or hospital, such as wandering in the hallways of the hospital or riding the bus to come participate in the experiment. *Sleep as performance* refers to any concern that one's sleep in the lab is inadequate, such as fear of disappointing the experimenter or anxiety over being unable to fall asleep or remember a dream. *Friends/Family in the lab* refers to family members, friends or acquaintances participating in the experiment or simply being present in the lab. *Object of observation* refers to feelings or knowledge about being observed or evaluated during one's sleep, such as feeling vulnerable as a participant, having one's brain waves scrutinized, experiencing a breach of intimacy, or abusive behavior from the experimenter.

**Temporal orientation.** The dream's temporal orientation was assessed for each LID: *Past* referred to lab-related events taking place in the past (e.g., coming to the lab by bus), *Immediate Future* referred to lab-related events taking place just after the sleep period was completed (e.g., false waking or awakening); *Near Future* referred to any post-lab activities that were continuous with the lab experience (e.g., going home after the experiment was over). Temporal orientation was not scored if dream activities could not be clearly situated in time. 31.7% of LIDs included projections to the future; specifically, 26.5% were projections to the Immediate Future (the dreamer is still in the lab) and 10.1% were projections to the Near Future (the dreamer left the lab environment). In contrast, only 2.6% included projections to the near past (e.g., coming to the lab)–the remaining dreams either possessed neither type or were ambiguous (66.7%) (Table 1).

**Lab incorporation themes differ between naps and overnight, but not between sleep stages.** As an exploratory analysis, we performed GLMM to assess if the occurrence of the 6

**Table 2. Examples for each of the 6 LID themes and 3 temporal orientation measures.**

| LID themes | Dream excerpts | Lab elements |
|---|---|---|
| **Sleep Performance** | I felt like maybe you (the experimenter) were disappointed in me or that I wasn't 'performing' well enough on the sleep side. [a] | People, Task |
| | I was worried by the fact that I absolutely had to fall asleep to pass the dream lab test. [a] | Place, Task |
| | I dreamt that I was laying right here and couldn't fall asleep. After a long time, someone came into the room to tell me the simulation was over–by then I felt too tired to wake up. [a] | Place, People |
| | And then I was with my parents and I was crying, because I hadn't succeeded in sleeping and I felt like very bad to have failed. [a] | Task |
| **Wayfinding** | I got lost wandering through the hospital, and found myself outside with all the gear still stuck to my face. Wandering through all the corridors trying to remember my way back. | Place, Object |
| | . . . I was doing the study and I had to go to the bathroom because I had to pee and on my way to the bathroom I took the wrong way back, so I was in the wrong hallways. [a] | Task, Place |
| | I left the room to find [the experimenter] or someone else, but no one I recognized was around. I walked into the hallway [. . .] I went back in the room that I was sleeping in and found another doorway off the far wall that led to another room the size of my bed. | Place, People |
| **Meta-dreaming** | At one point I realized that I didn't have access to anything in the bedroom, so that I was probably dreaming, I became aware of my body which was very tense and immobile in the bed. So I took control of my dream, I started walking a few steps in the hospital, but my legs were very heavy. . . [a] | Place |
| | I look at my hands and see that I am sleeping. I 'wake up'. I am in the lab and I wake up from my nap. I don't really know if I'm awake or asleep. [a] | Place |
| | I turn off the TV, and tell myself I should fall asleep to have a dream for the study I'm participating in. I don't move, and eventually I feel vibration in my whole body. I also feel that I'll start levitating in the air. [. . .] I'm happy to be able to have a lucid dream during the study. But because of my anticipation, I couldn't enter in a dream (in my dream). | Task |
| | I was in my father's living room and I absolutely had to fall asleep. I didn't have electrodes on me [. . .]. I dozed off on the couch. [. . .] [My sister] told me a woman had slept on the couch to examine me. [. . .] She was sorry she fell asleep and explained that we'd have to start all over again, because she slept for too long. At this point, I didn't understand what was going on and then I remembered that I was already doing an experiment with the electrodes and everything. [a] | Sleep, People, Object |
| **False awakening–False waking** | I dreamed that I had failed to sleep here. I had spent the whole night not sleeping. [. . .] I left around 6am, we sort of gave up, and then I left, but not to my home but to my parents. [a] | Place |
| | I dreamed of waking up here in the lab. I woke up and [the experimenter] explained to me that it was over and I could go home. [a] | Place, People |
| | Then I thought I was awake and I was waiting for the end of the experiment, I glued one of my electrodes back on. [a] | Place, Object |
| | I dreamt that I knocked on the door because I was done sleeping to let [the experimenter] know, and she said she will be here soon. | Place, People |
| **Friends/Family in lab** | . . . doing this study only slightly different, a friend came with me and was hitting all the buttons,. . .and they got yelled at. | Place, Task |
| | When I got to the other room, my mother and my sister were there, I thought that was normal. I was very hungry, my mother who worked in the lab did not want to feed me. . . [a] | Place |
| | . . . doing the sleep experiments but actually my parents were in my dreams doing the same experiments with me. But I did not have any electrodes and while they were in my bed, I was on an inflatable mattress near them. [. . .] I was explaining about this sleep experiment to my parents and that they needed to behave because we were being recorded. | Place, Task |
| | . . . I was with 3 friends in the lab, filling out questionnaires. [a] | Place, Task |
| **Sensory** | . . .there was someone on top of me in the lab while I was in bed, but I was paralyzed, then the person was pushing to hold me against the bed, then I tried to scream, then I was moving my eyes, then I tried to move and I started screaming [. . .] I felt like the person in black had their arms around my head or close to my headboard. [a] | Place |
| | I was sleeping, then I was hungry and also wanted to go to the bathroom so I left the room . . . [a] | Place |
| | A friend of mine takes a water gun and tries to get the [electrode] paste out of my hair . . . [a] | Object |
| | I woke up in the room where I was taking a nap [. . .] I was sitting on the floor because I couldn't support my weight anymore, I was completely heavy and exhausted. The assistant went into the other room by holding one end of the wires, pulling so hard that I couldn't follow her, they were tearing from my skin. . . [a] | Place, Object |

*(Continued)*

**Table 2.** (Continued)

| LID themes | Dream excerpts | Lab elements |
|---|---|---|
| **Object of observation** | . . . I was in a glass room, and I was hooked up to electrodes. . . | Object |
| | . . . exactly the same as here except that there were windows instead of walls. [a] | Place |
| | The researcher told me I had something very weird with my brain. | People |
| | I did a test where she put electrolytes in my head, and it was supposed to write the dreams directly on paper. [a] | Task |
| | I was trying to fall asleep in my dream by holding onto an office chair and rocking it back and forth, aware of the cameras but desperate enough to fall asleep that I didn't really care. | Place, Object |
| | . . . strangely this room was not private. Even though in my dream only me and my family were present, I had the feeling that other people I don't know were present, without seeing them. [a] | Place |
| **Temporal orientation–Past** | I had a dream about coming to the hospital by bike. [a] | Place |
| | I was rushed to get out, probably I wanted to come to the lab to do the experiment, but I'm not sure [. . .] after that I rushed to get the bus, I remember I was worried since apparently the bus was late, I was afraid to come late. | Place, Task |
| | I was coming for the project. . . the project here and. . . to sleep. [a] | Place, Task |
| **Temporal orientation– Immediate Future** | . . . there are a lot more people than when I fell asleep. I look at the time, it is ten past noon. [a] | Place |
| | Actually, I was dreaming that I was here and that I was waking up. After that, we were leaving the room, you were taking the electrodes off me. [a] | Place, People, Object |
| | I was sitting in something like a laundry basket and leaning on her chair trying to sleep. I looked outside and it was night already so I wondered why we were here so late. | Place, People |
| **Temporal orientation–Near Future** | . . . this was after a false awakening where I thought I returned home and my roommate took my blankets because he was getting too cold. | Object |
| | I was then walking with my colleagues at McGill talking about my experience in this study. | Task |
| | . . . finally the sun was setting and suddenly it was dark, and I was walking at that moment. I had. . . the electrodes. . . I was telling them that I was at an experiment so I looked and then I thought: "Oh they forgot an electrode". [a] | Object, Task |

[a] translated from French.

specific LID themes could be predicted by Sleep stage, Sleep timing or Nightmare proneness (significance level at p = .05/6 = .0083). Sleep timing (naps>overnight) only predicted the occurrence of Meta-dreaming (p = .003) and, more marginally, Wayfinding (p = .025) and Sensory (p = .011), but not Sleep performance (p = .875), Friends/Family in lab (p = .199) or Object of observation (p = .945). Sleep stage (all p>.229) and Nightmare proneness (all p>.07) did not predict any of the LID themes (see Table 1 for percentages of LID themes by Sleep stage and Sleep timing).

**Table 3. Exemplary lab incorporation dream in which 5 different themes were identified.**

Global score: 7; Sleep stage: N2 awakening; Sleep timing: nap; original language: French
Themes: Meta-dreaming, Wayfinding, Sensory incorporation, Friends/Family in lab, Sleep performance

*I see the dream in a circle in the center of my vision. In the rest of my field of vision I think I can **see the laboratory room**. I **see the wires on my face**. I **try to go back to sleep**. I fall back into the dream several times, and just as I wonder if I'm dreaming and if it's a **lucid dream**, I pop out of it, and it just becomes a circle in the center of my field of vision. I am conscious of my body again and I have no dream sensations. But I can control it. There are **hospital corridors that I hover backwards through**. The **student researcher** follows me and I seem to want to escape her. I sneak into corners, wardrobes and ventilation ducts, always fluidly and backwards. I feel fun. **I open a door**. Commercial door. Maybe **the ones from the hospital**. I can feel the cold metal part for pushing. I go out into the street. An avenue. [. . .dream scenes out of lab. . .]. Right then I can **feel my body in bed** and I'm not sure if I'm dreaming. Then **I'm in the lab**, I'm sitting on the floor in the **virtual reality room**. In the rectangle. **My parents come** into the room. I am annoyed. **I have to concentrate, I only have a few minutes to fall asleep, I have to sleep 90 minutes!** I tell them. I have to limit distractions, I have to concentrate. My father leaves but **my mother sits** on the floor with me, in the rectangle. I don't understand what she is doing there. The hospital turned into a kaleidoscopic structure that I fell into. Geometric holes. I fell into the lens of a camera and **saw myself** going through each of the lenses. I had no body, I was a point. There was a lot of rapid transformation of the dream. Abstract volumes. A soft **computer keyboard** that **slides between two blankets**. Shapes, such as a sphere, having the texture of water in the ocean, but black. Breaches from which light sprang. With each transformation or evolution of forms, there were associated sounds. I imagined myself in my **body superimposed on the one in the bed**, but **I could just barely move the virtual body through the real one that was not moving**.*

We also assessed whether future temporal orientation (both Immediate and Near Future) varied by time of night, using both Clock time of awakening and Sleep stage as predictors. Future temporal orientation was predicted by Clock time of awakening, being more likely in dreams recalled later in the morning ($\beta$ = 0.271, SE = 0.112, z = 2.440, p = .0147), but was not predicted by Sleep stage ($\beta$ = -0.648, SE = 0.112, z = 2.440, p = .151).

## Discussion

The results show that dreaming of the lab occurs in over a third of all dreams collected in the lab, and in almost half of participants who succeed in recalling a dream. Moreover, LIDs are much more likely in the morning, occurring in over half of dreams collected from morning nap REM awakenings, than they are in overnight dreams. These quantitative results thus point to two main physiological pressures–sleep stage and sleep timing–to which lab incorporation dreams are sensitive. Our qualitative analyses further identified six narrative themes associated with lab incorporation dreams, including *Meta-dreaming*, such as lucid dreams and false awakenings, *Sensory incorporations*, *Wayfinding* to, from or within the lab, *Sleep as performance*, *Friends/Family in the lab* and *Being an object of observation*. These themes point to possible pressures driving dream formation that include: social affiliation, skill rehearsal, self-projection in space and time, and monitoring of one's current state.

Considering the lab as a shared memory source over several standard protocols has thus enabled us to clarify 1) when recent episodic memories are selected for dream replay (pointing to physiological pressures that may influence dream reactivation); and 2) how diverse simulations are created from episodic memory fragments (pointing to cognitive pressures that may structure the formation of dream narratives).

### Lab incorporations in dreams are common

Our estimated prevalence of LIDs (35.8%) is in line with previous studies which report, on average, that 38.4% (range: 32.2% to 68.0%) of dreams refer directly or indirectly to the experimental setting [2]. Because we did not distinguish between direct and indirect lab references in defining the SoLID criteria, these criteria captured most types of indirect references, including general occurrences of laboratories, elements related to sleep or dreaming, or participating in an experiment (see [2]).

In addition to replicating previous results, we add several new findings. First, incorporations of lab memory elements are especially high for dreams that were collected from either REM sleep (44.2%), morning naps (48.4%) or, especially, REM sleep from morning naps (55.2%). Second, not only is the prevalence of LIDs increased in morning naps, but so too is their richness. Third, participant factors have very limited relationships to LID prevalence. Neither nightmare-proneness, sex nor anxiety levels are related to LIDs; only depression levels marginally predict their presence. Fourth, we did not find a general effect of sleep manipulation on the occurrence of LIDs. Finally, LIDs differ from non-LIDs in being more clearly recalled, more confusing and less emotionally positive.

**Time-dependent occurrence of memory processes in dreams.**   Our finding of high levels of LIDs from morning naps is open to a number of possible explanations. One possibility concerns the time-dependence of memory processing. Memory reactivations during sleep are considered to be a time-sensitive component of memory consolidation that may also implicate dream reactivations of specific memory contents [25, 38, 39]. Day-residue memory sources are known to be common in dreams (65–70%, see review in [18]) and in the present context may have favored reactivations of lab memories that occurred almost exclusively immediately prior to sleep. This might especially be the case for morning naps, during which the repertoire of

'previous day' elements was limited in large part only to travel to the lab and the lab experience itself. Moreover, memory encoding has been found to be more efficient in the morning than later at night, possibly due to circadian fluctuations in levels of arousal [40], which could in turn increase the incorporation rates of pre-sleep events in morning nap dreams. On the other hand, our finding that later awakening times predict higher rates of incorporation, although possibly confounded by the type of protocol used (overnight vs nap), is contrary to evidence that early night dreams more frequently incorporate recent experiences than do late night dreams [41, 42] and that memories early in the sleep period are more episodic than semantic [43]. In fact, previous overnight studies found that LIDs tend to occur more often in early nighttime REM periods [11, 44], whereas others found no time of night effect [5, 6]. Nonetheless, these effects may not be due to circadian influences, but to early night dreams arising sooner after prior wakefulness than late night dreams, a situation that is also the case for our morning nap dreams compared with overnight dreams.

**Lab incorporation dreaming as a hyper-aroused state?.** A second possible explanation of our high LID prevalence for morning REM naps is that these dreams are influenced by higher levels of arousal. Heightened cortical arousal associated with circadian/ultradian factors or with sleeping in a novel environment—or both—may induce a particularly vigilant sleep, with increased processing of lab-related sensory or environmental stimuli and, critically, the incorporation of some of these stimuli into dream content.

A first possible source of arousal is the physiological state of morning nap REM sleep, which is a more cortically aroused state than either early night REM or NREM sleep [45, 46]. We find that while LIDs can occur in any stage of sleep, they are twice as frequent in REM as in NREM sleep, independent of time of night. Similarly, the proportion of dreams incorporating lab personnel was previously found to be higher in late REM dreams than in NREM dreams [47]. Here, we find that the nature of this REM-NREM difference in LIDs is more a quantitative than a qualitative one, as the LIDs from the two sleep stages do not differ in other dream attributes, such as emotional intensity, or in specific LID themes. The specific case of false awakenings, which were very frequent in our study, is also consistent with the suggestion that LIDs are influenced by heightened REM sleep arousal. False awakenings were suggested to be a hyper-aroused REM sleep state [48], as recently evidenced by spectral EEG analysis [49]. Moreover, false awakenings frequently occur in association with lucid dreams [48, 50–53] and sleep paralysis [50, 53] both of which have also been proposed to be highly aroused, hybrid states with features of both REM sleep and waking [54, 55].

A second source of arousal that may have affected LID prevalence in our study is the stimulating effect of sleeping in a novel environment—or the first-night effect. The first-night effect generally involves lighter sleep, prolonged sleep-onset latency, lowered sleep efficiency and increased intermittent wake time—all features consistent with increased arousal [56, 57]. Changes in dream content have also been reported [12]. Tamaki et al. [58] report an interhemispheric asymmetry in NREM sleep depth during first night sleep, by which one hemisphere—even though asleep—appears to be more aroused and able to process external information to a greater degree than the other. They also found increased brain responses to external stimuli in both hemispheres during phasic REM sleep during a first night in the lab [59]. Indeed, the sleep lab is replete with novel stimuli that could contribute to first-night arousal: participants find themselves in an unfamiliar building, bedroom and sleep routine, with electrodes stuck to their head and body, and experiencing unwanted body sensations such as hunger, cold, or having to urinate. While in the present study we could not directly address the first-night effect, as most participants only visited the laboratory once, some studies have shown that LIDs occur more frequently during a first night in the laboratory [11, 12], and others that they persist even after sleeping multiple nights [5, 6, 10]. Morning nap

protocols may even have amplified first-night effects relative to overnight sleep protocols, since participants in nap studies have much less time to acclimate to the laboratory.

In sum, potential explanations for the high prevalence of lab incorporations into dreams include the physiological effects of 1) time-dependent factors governing memory processing in sleep, such as day residues in morning naps, and 2) factors increasing arousal during sleep, including morning REM sleep cortical arousal and the first-night effect.

## From memory fragments to world simulations

Our qualitative analysis revealed that some fundamental elements of episodic lab memories consistently appear in dream content especially, in decreasing order of prevalence, places, people, tasks, objects and sleep-related activities. The analysis further provides evidence that although dreaming frequently reactivates these elements, it does not replay them in their episodic entirety. In fact, episodic reconstructions in dreams are almost always incomplete [60]. One view of the 'incompleteness' of episodic replays is that dreaming operates beyond the episodic memory system, 'breaking down' waking memories and recombining them into novel dream scenarios [61]. From our assessment of the present data set, we contend that there are certain design pressures which organize fragments of recent experience into dream scenarios that are preferentially—at least in the case of LIDs—of four distinct, albeit inter-related, types: social, skillful, self-projectional and state-monitoring (Fig 2). We consider each of these hypothesized pressures in the following sections.

**Social.** Next to depictions of the lab setting itself (75% of LIDs), lab-related people were the most (55%) commonly incorporated element, and 16% of LIDs featured friends and family members present in the context of the lab. That dreamed lab simulations are preferentially social is not unique; social situations are found in 83.5% of all dream reports collected in a home setting [61]. The social dimensions of visiting the sleep lab are undeniable: it is an intimate interpersonal encounter in which a participant's performance and private dreams are not

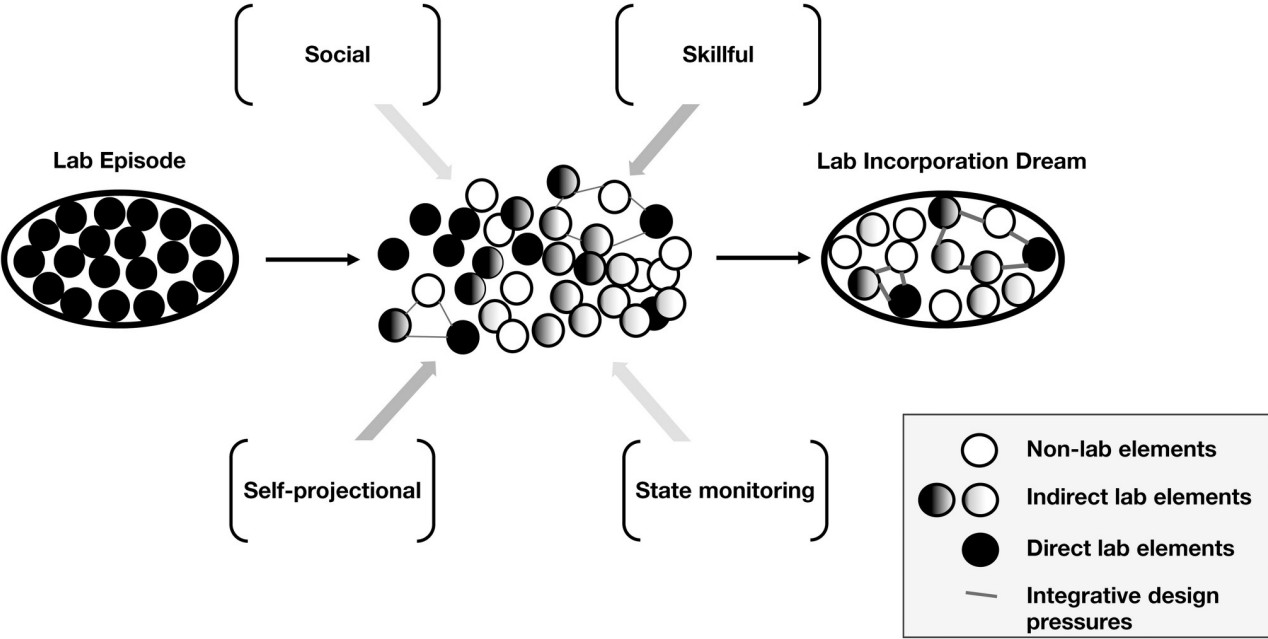

**Fig 2. Construction of lab incorporation dreams from episodic memory fragments under integrative design pressures.**

only observed but objectively scrutinized and compared to those of other participants. Other authors have remarked that hospital personnel are likely to trigger anxiety among lab volunteers, being perceived in their dreams as '. . .sadistic, incompetent and assaultive people. . .' [5]. This may underlie the awareness of being an object of observation, which frequently accompanies incorporation of the experimenter or other observers into LIDs. A similar theme was discussed in previous studies in which lab-related contents in dreams were perceived as 'threatening and embarrassing' [10], involved 'harm or discomfort to the subject' [8], or contained anxious responses toward the experimenter, toward the experimental room, or concerns about the participant's own behavior [5]. In contrast, it has been suggested that a pleasant lab experience with warm and friendly staff could diminish the first-night effect [12, 62], indicating further that the nature of social interactions within the lab influence how participants sleep and dream.

To the extent that friends and, especially, close family members are associated with comfort and support in relation to sleep (e.g., early memories of being tucked into bed and watched over by caring parents; falling asleep with a bed partner, etc.), their presence within dream scenarios in the lab bedroom may reflect associations to the participants' very recent experience of being put to bed and watched over by the experimenters.

**Skillful.** That dream design pressures are frequently skillful and performative in nature, i.e., fashioned around a theme of skill rehearsal, is reflected in many of the LIDs in our data set. LIDs often incorporate basic tasks that are part of most sleep studies, including answering questionnaires, trying to remember dreams or simply 'being part of a study' (54%), almost as often as they incorporate social features. These findings align with theories and evidence that dreaming participates in functions of learning, rehearsal and long-term memory consolidation [39, 63–65]. Accordingly, one frequently discussed possibility is that dreaming about recently learned information or skills is an integral part of the memory reactivations thought to underlie sleep-dependent memory consolidation [25, 39, 64].

Beyond specific controlled tasks that are part of a study protocol, many of the seemingly mundane activities of a sleep lab may be experienced rather as skill-demanding challenges from a participant's perspective. Merely falling asleep 'on command' is a challenge for many, as is the notion of 'remembering to recall a dream' when awakened unexpectedly, the completion of questionnaires and the performance of other tasks by computer keyboard. Similarly, for some, the task of navigating to, from and within a novel environment may trigger a dream rehearsal of spatial skills, likely reflected by the high frequency of wayfinding scenarios. In sum, skillful design pressures may be at work in LIDs for which salient sleep lab tasks—or task components—have been prioritized for longer-term memory consolidation.

**Self-projectional.** Projections of the dreamed self through space and through time are common dream structuring mechanisms that also appear frequently in our dream set. In LIDs, a dreamer typically explores the surrounding sleeping environment or anticipates the unfolding of a post-awakening scenario. Wayfinding dream themes, in particular, may reflect the hippocampally-dependent sleep function of consolidating and facilitating spatial memories. Actively navigating the dream world may, in this view, subserve a broader spatial memory function by which participants dream about important spatial and navigational features of the lab in order to (implicitly) establish new familiar landmarks, environmental anchors or critical routes for future use. Our observations that participants' dream worlds often involve self-projection into potential future scenarios (32%), which were much more frequent than projections into the past (3%), also aligns with the purported hippocampal function of future imagining or mental time travel more generally [66]. There occurred frequent instances during which participants dreamt of waking up, of being awakened or of being awake, of reporting their dreams, and even of leaving the lab after the experiment had finished. These future-oriented episodes

also frequently included spatial-exploratory behavior in and around the sleep lab—possibly pointing to the simultaneous activation of both spatial and temporal hippocampal processes. Such concatenations of temporal and spatial features were previously identified as one typical scenario characterizing false awakenings [48] and further supports the implication of hippocampal organizing pressures in dream formation. Other research showing that stimulus-independent mentation, such as dreaming and mind-wandering, is especially likely to be about the future [67, 68] is consistent with the purported prospective or future-oriented function attributed to offline states [69–73]. In wake, the ability to simulate and anticipate future narratives builds on the ability to remember and use episodic memories [74–77]. LID narratives, too, incorporate episodic memory fragments for the construction of possible future scenarios about the ongoing lab experience, potentially preparing the dreamer for both perception and action in subsequent wakefulness.

**State-monitoring.** The frequent appearance in our LID sample of themes expressing participants' concerns with, and awareness of, ongoing bodily and environmental sensations suggests the operation of a design pressure to monitor and update an individual's model of themselves and input from their surroundings while they dream. These themes include minimal awareness of ongoing sensory experiences (sensory incorporation, sleep-related activities), awareness of the true sleeping environment and context (being an object of observation), and awareness of alterations in one's state of consciousness (meta-dreaming). In our sample of LIDs, it was common for participants to dream about waking up with a need to use the bathroom or to retrieve food; they dreamt of being cold, being paralyzed or weak, or feeling the electrodes on their scalp or skin. Such scenarios relate not only to known past events, but also to ongoing sensations that are potentially instantly available to consciousness during sleep. In addition, some dreams appear to incorporate sleep sensations themselves, e.g., dreamers suddenly fainted or fell asleep in the dream; these sometimes occurred as activities outside of the lab but were nonetheless remarkably akin to elements of the ongoing lab experience. Moreover, dreamers seemed to remain aware of the broader context of being under close scrutiny by the experimenters; LIDs commonly displayed scenarios in which cameras were present, the bedroom walls had become windows, participants' thoughts were exposed, or their intimacy was compromised. Such instances of ongoing self-monitoring and monitoring of the environment again align with the idea of an increased vigilance, or 'night watch', emerging when sleeping in an unfamiliar environment [58]. Finally, beyond the monitoring of bodily sensations and of the environment, participants' LIDs suggested that they maintained a variable degree of self-awareness during dreaming that manifested in a variety of changes in their state of consciousness, i.e., in the frequent occurrence of lucid dreams, false awakenings, sleep paralysis and so forth.

## Methodological and fundamental implications of LIDs

Our findings have implications for fundamental dream research including both methodological advantages and limitations. Lab dream reports can be collected and compared across several studies and even across independent laboratories, enabling the examination of larger dream data sets all influenced by a common and easily identifiable presleep experience.

However, LIDs constitute a clear instance in which the scientific act of observation influences the desired object of measurement [2, 10]. Others have expressed caution that dreams collected in sleep lab environments are not ecologically valid proxies for natural dream content [78, 79] and, more generally, that sleep behavior and quality often differ substantially when measured in the lab compared to the home sleep environment. For example, lab dreams are less emotional than are home dreams [80] and nightmares seldom occur in the lab setting [81,

82]. Cases where participants were aware of being observed and recorded, or self-aware of their thoughts while dreaming illustrate how sleeping in a lab may alter or censor ongoing dream content. Although the high frequency of LIDs might seem to hinder dream theme diversity in that their high prevalence in the lab is arguably at the expense of other dream themes, our qualitative analysis reveals that even within LIDs a diversity of themes is apparent. In fact, most of the identified design pressures appear to reflect underlying, interrelated functions attributed to dreaming, such as social simulation, memory consolidation, wayfinding and map-building, preparation for future perception/action and maintaining an awareness of the self and the environment.

Another problem that may occur in the case of LIDs is that participants are at times uncertain whether realistic laboratory dreams had rather actually occurred during wake. Specifically, participants may experience 'false waking' dreams in which they report not having dreamt or not having been asleep, despite physiologically verified sleep or phenomenologically verified dreaming (e.g., reports of events that never happened, such as 'you called my name and told me the experiment was over'). This possible confusion between dreaming and waking realities, that seems especially common during lab sleep, has important implications for studies assessing the neural correlates of dream recall, where the presence or absence of conscious experiences is a central variable of interest. Informing participants prior to sleep on the variety of LIDs they may experience in the lab could help them better identify those 'false wakefulness' episodes as dream experiences. LIDs may also provide a useful avenue for the study of related sleep phenomena such as false awakenings, out-of-body experiences, sleep paralysis and lucid dreaming. For example, participants could be trained to perform reality checks whenever they notice particular elements of the lab experience (and perhaps particularly the themes described here), which in turn could help them become lucid in a LID.

Finally, while LIDs may inform our understanding of memory processes related to dreaming, the lab experience may also compete with other tasks being investigated; the lab experience may even be prioritized above other experiences, including experimental learning tasks that participants complete before sleep. In line with this, recent studies have found that, within a single protocol, lab-incorporations and specific task-incorporations rarely occur within one and the same dream [3, 25]. This evidence suggests that incorporation of the lab into dreams may interfere with the desired incorporation of more targeted learning tasks. Despite these limitations, analyzing LIDs in greater detail may help researchers to develop learning tasks that successfully compete in importance with the lab experience, e.g., tasks that have a strong social component or that require wayfinding. This may help rectify the common problem of so few incorporations of a learning task appearing in dream content that dreaming's functional role in that task cannot be properly assessed [83, 84]. Moreover, the main design pressures identified in the LIDs from the present data set may help in explaining patterns of dream formation and dreaming's relationships with episodic memory more generally. Assessing the extent to which non-LIDs are structured by design pressures similar to those identified for LIDs would further clarify the generalizability of our findings.

## Limitations

A number of limiting factors are present in this study. First, the basic episodic elements (people, objects, places, tasks and sleep-related activities) on which we based our scoring are not exhaustive; for example, we expect emotions to be another important component of episodic memory that is likely reflected in dreams. Because we did not measure the pre-sleep emotional response of our participants when they came to the laboratory, we could only speculate on the influence it had on the construction of LIDs.

Second, we could not control the influence of different types of sleep manipulations on LIDs due to convergence problems in our statistical model. However, we were able to show that lab incorporation rates do not differ between control participants (whose sleep was not disturbed) and participants in experimental situations (TMR, tACS, etc.). This does not fully preclude that a specific type of stimulation may have led to more lab incorporations but does confirm that lab incorporations in general are not dependent on experimental stimulations. Further studies could address how different stimulations during sleep, especially those leading to cortical arousal, may affect LIDs.

Third, while we showed that lab incorporations can occur in any sleep stage, lower sample sizes for N1, N2 and N3 awakenings prevented us from statistically assessing changes in incorporation rates across levels of NREM sleep depth.

Fourth, perhaps because our lab is located in a labyrinthine hospital situated far from the city center and participants typically need detailed instructions to find it, navigation or wayfinding themes may have proven to be more common in our dream set than in other studies. While this specific situation may have biased the proportion of LIDs containing wayfinding scenarios, it may nonetheless have highlighted an aspect of dreaming that is otherwise often overlooked.

Fifth, 28% of LIDs in our study did not correspond to any of the six identified themes. Most of these dreams were unique scenarios taking place outside the laboratory setting, while incorporating one or more isolated aspects of the lab, such as meeting the experimenter outside of the lab, or noticing the electrodes in some other context. These LIDs may nonetheless be structured by similar overarching design pressures such as social affiliation and state-monitoring.

Finally, it is possible that additional organizing themes were present in the LIDs, but were simply not identified in our qualitative assessment. Further studies looking at LIDs reported in different laboratories would help clarify this question.

## Supporting information

**S1 Appendix. Scoring of laboratory incorporation dreams (SoLID) criteria.**
(PDF)

**S2 Appendix. Exemplary laboratory incorporation dreams.**
(PDF)

## Acknowledgments

The authors sincerely thank Tyna Paquette for help with coordination of all the studies involved, Arnaud Samson-Richer for help on dream scoring, Maude Pastor for help with the database preparation and all people involved in collection of the dreams throughout the years.

## Author Contributions

**Conceptualization:** Claudia Picard-Deland, Tore Nielsen, Michelle Carr.

**Formal analysis:** Claudia Picard-Deland, Tore Nielsen, Michelle Carr.

**Funding acquisition:** Tore Nielsen, Michelle Carr.

**Methodology:** Claudia Picard-Deland, Tore Nielsen, Michelle Carr.

**Supervision:** Tore Nielsen.

**Visualization:** Claudia Picard-Deland.

**Writing – original draft:** Claudia Picard-Deland, Tore Nielsen, Michelle Carr.

**Writing – review & editing:** Claudia Picard-Deland, Tore Nielsen, Michelle Carr.

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
