## [Decision Letter · Decision Letter 0]

16 Aug 2021

PONE-D-21-23305

Dreaming of the sleep lab

PLOS ONE

Dear Dr. Nielsen,

Thank you for submitting your manuscript to PLOS ONE. After careful consideration, we feel that it has merit but does not fully meet PLOS ONE’s publication criteria as it currently stands. Therefore, we invite you to submit a revised version of the manuscript that addresses the points raised during the review process.

I request that you make major revisions before it is processed further. Please carefully consider all issues mentioned in the reviewers' comments. In particular, I reccomend you to focus on these aspects:

clarification of some methodological aspectsre-organization of the discussion addition of some limitations

We look forward to receiving your revised manuscript.

Kind regards,

Serena Scarpelli

Academic Editor

PLOS ONE

Journal Requirements:

2. Please modify the title to ensure that it is meeting PLOS’ guidelines (https://journals.plos.org/plosone/s/submission-guidelines#loc-title). In particular, the title should be "specific, descriptive, concise, and comprehensible to readers outside the field" and in this case it is not informative and specific about your study's scope and methodology.

Reviewers' comments:

Reviewer's Responses to Questions

**Comments to the Author**

1. Is the manuscript technically sound, and do the data support the conclusions?

Reviewer #1: Yes

Reviewer #2: Yes

2. Has the statistical analysis been performed appropriately and rigorously? 

Reviewer #1: Yes

Reviewer #2: Yes

3. Have the authors made all data underlying the findings in their manuscript fully available?

Reviewer #1: No

Reviewer #2: No

4. Is the manuscript presented in an intelligible fashion and written in standard English?

Reviewer #1: Yes

Reviewer #2: Yes

5. Review Comments to the Author

Reviewer #1: In this work, Picard-Deland and colleagues investigated the incidence and features of laboratory incorporations into dreams. Moreover, they assessed the possible relationship between incorporations and individual psychological traits. They found that incorporations 1) occurred in about a third of all dreams; 2) occurred more often for REM sleep and during morning naps; 3) are minimally or no influenced by participant factors; 4) were more clearly recalled but also more subjectively confusing; 5) were found to relate to one or more common themes.

The study results extend and complement previously published results on the same topic, bringing in novel findings and interesting observations. Taken as a whole, I have a positive opinion of this manuscript. However, some information should be added regarding the methodology, and some limitations should be more thoroughly discussed. The Discussion section is lengthy and could be shortened. Below are my detailed comments and suggestions for the authors.

- Line 33. The abstract seems to imply that incorporations were more common only in REM from morning naps, but results actually indicate that stage and timing are independent predictors.

- Line 163. The authors regrouped dream reports obtained from N1, N2, and N3 as NREM. While this was necessary to avoid excessively unbalanced sample sizes across stages, it also limited the possibility to detect possible differences in incorporation rate across levels of NREM 'depth'. This choice and its limitations should be briefly discussed.

- Lines 169-171. Please clarify if sleep restriction or deprivation protocols (or any other procedures aimed at favoring sleep) were used during morning nap studies.

- Lines 172-176. Please clarify if these scores were available from both sleep-manipulation and no-manipulation protocols and, in the latter case if sleep-manipulation was controlled for in analyses investigating the relationship between incorporation rate and individual factors.

- Lines 182-188. The described protocols appear quite different from each other. For instance, some protocols involved applying on the participant additional instruments (e.g., tDCS electrodes or pressure cuff), while other protocols involved manipulations that may not have been subjectively perceived as "intrusive" by the volunteers (e.g., TMR). I am relatively skeptical regarding their merging under the same “sleep manipulation” category. If a more in-depth characterization of the impact of different protocols on incorporation is not possible, this limitation should be adequately discussed.

- Line 196. If my understanding is correct, two of the raters are student (J2) and supervisor (J3). In this respect, even if they rated the reports “independently,” they may not constitute fully "independent” raters. This may explain why the inter-rater agreement was very high between these raters for most categories. Please include in table or text the level of inter-rater agreement with respect to the third rater, J4, and comment on relevant differences (if present) between J2/J3 scores and J4 scores.

- Lines 432-434. The authors may consider as explanation also potential differences in memory encoding between night and day experiments related to the level of arousal during the pre-sleep phase (e.g., Baddeley et al., 2007, Memory and Time of Day), rather than during actual sleep as they suggest (Lines 445-449).

- The Discussion section is extremely long and almost constitutes a distinct article. I would suggest to the authors to move parts of the Discussion - particularly the very long section regarding the themes of incorporation and their interpretation - in supplementary material and only include in the manuscript a shortened summary of the most relevant aspects. This would improve the overall readability of the manuscript.

Reviewer #2: This study considered 528 dreams of 343of participants) collected in a Montreal sleep lab aiming to investigate laboratory incorporations using a new scoring system.

As main quantititve findings lab incorporations occurred in over a third of analyzed dreams and were mainly associated to REM sleep and morning naps.

Concerning qualtitive aspects, the study shows that themes associated with lab incorporation were: Meta-dreaming (e.g., lucid dreams and false awakenings, Sensory incorporations (27%), Wayfinding, Sleep as performance, Friends/Family in the lab and Being an object of observation. A reletively high proportion of dreams contained more dreams incorporation of projections into a near

future than into the past

On the whole, I have a postitive opinion on this study which provides relevant pieces of knowledge for studying dream recall in laboratory settings (i..e, a methodological merit), but also for increasing knowledge on the sources of dream per se.

I have only mionr points:

1. Sample size is strength and weakness of this study, since many sources of variability have been included for collecting such sample: I am not completely convinced that the statstical design was successful in controlling all the sources of variability

2. Percentages of recall from NREM (64.3%) were quite lower compared to other studies: On the other hand, percentages upon awakenings from N3 seem unusually high (64.3%). In my opinion, the authors should make an effort to explain this finding and to discuss it in relation to previous studies.

3. Concerning the main finding related to the effect of sleep stage and morning naps, I am not completely convinced that performing Sleep stage or Sleep timing as the only predictor is sufficient to exclud an interaction between these basic factors. This issue should be discussed.

4. The analyses also are not completely convincing in excluding that experimentally induced sleep arousal alone does not cause LIDs. Cortical arousal is a main determeint of the probability to have dream recall, and (different) experimental manipulations may have affected LIDs.

6. PLOS authors have the option to publish the peer review history of their article (what does this mean?). If published, this will include your full peer review and any attached files.

Reviewer #1: No

Reviewer #2: **Yes: **Luigi De Gennaro

---

## [Author Response · Author response to Decision Letter 0]

3 Sep 2021

Reviewer #1

--General comment

In this work, Picard-Deland and colleagues investigated the incidence and features of laboratory incorporations into dreams. Moreover, they assessed the possible relationship between incorporations and individual psychological traits. They found that incorporations 1) occurred in about a third of all dreams; 2) occurred more often for REM sleep and during morning naps; 3) are minimally or no influenced by participant factors; 4) were more clearly recalled but also more subjectively confusing; 5) were found to relate to one or more common themes. The study results extend and complement previously published results on the same topic, bringing in novel findings and interesting observations. Taken as a whole, I have a positive opinion of this manuscript. However, some information should be added regarding the methodology, and some limitations should be more thoroughly discussed. The Discussion section is lengthy and could be shortened. Below are my detailed comments and suggestions for the authors.

--Comment 1

Line 33. The abstract seems to imply that incorporations were more common only in REM from morning naps, but results actually indicate that stage and timing are independent predictors.

--Reply 1

We clarified this distinction in the abstract, as follow (line 33, additions in bold):

“Lab incorporations occurred in over a third (35.8%) of all dreams and were especially likely to occur in REM sleep (44.2%) or from morning naps (48.4%).”

--Comment 2

Line 163. The authors regrouped dream reports obtained from N1, N2, and N3 as NREM. While this was necessary to avoid excessively unbalanced sample sizes across stages, it also limited the possibility to detect possible differences in incorporation rate across levels of NREM 'depth'. This choice and its limitations should be briefly discussed.

--Reply 2

We already report incorporation rates separately for N1, N2 and N3 on line 249: “They occurred in all stages of sleep: N1 (17/104; 16.4%), N2 (21/78; 26.9%), N3 (2/9; 22.2%), NREM combined (40/191; 20.9%), REM (149/337; 44.2%).” 

However, low sample sizes prevented us from looking at statistical differences. We clarified our choice in the Methods section as follow (lines 166-168, addition in bold):

“In order to avoid sample sizes that differ too much across sleep stages and because we had no specific hypotheses about dream incorporation as a function of NREM sleep depth, reports from N1, N2, and N3 awakenings were regrouped as NREM awakenings (N=277).”

We also now discuss this limitation in the Discussion (lines 790-92):

“Third, while we showed that lab incorporations can occur in any sleep stage, lower sample sizes for N1, N2 and N3 awakenings prevented us from statistically assessing changes in incorporation rates across levels of NREM sleep depth.”

--Comment 3

Lines 169-171. Please clarify if sleep restriction or deprivation protocols (or any other procedures aimed at favoring sleep) were used during morning nap studies. In the latter case if sleep-manipulation was controlled for in analyses investigating the relationship between incorporation rate and individual factors.

--Reply 3

Sleep restriction, deprivation or facilitation protocols were not used for any of the nap studies so no statistical controls were implemented.

--Comment 4

Lines 182-188. The described protocols appear quite different from each other. For instance, some protocols involved applying on the participant additional instruments (e.g., tDCS electrodes or pressure cuff), while other protocols involved manipulations that may not have been subjectively perceived as "intrusive" by the volunteers (e.g., TMR). I am relatively skeptical regarding their merging under the same “sleep manipulation” category. If a more in-depth characterization of the impact of different protocols on incorporation is not possible, this limitation should be adequately discussed.

--Reply 4

We agree that there is variability under the ‘sleep manipulation’ category, but the GLM models we used could not control for all different types of manipulations without leading to convergence problems. Our model however shows that lab incorporation rates do not differ between control subjects (who were without any type of interference during sleep) and subjects in experimental situations (TMR, tACS, etc.). This does not fully preclude that a specific type of stimulation may have led to more lab incorporations but confirms that lab incorporations in general are not dependent on most types of experimental stimulations.

We now further discuss this limitation in the Discussion (lines 808-815):

“Second, we could not control for the influence of different types of sleep manipulations on LIDs due to convergence problems in our statistical model. However, we were able to show that lab incorporation rates do not differ between control participants (whose sleep was not disturbed) and participants in experimental situations (TMR, tACS, etc.). This does not fully preclude that a specific type of stimulation may have led to more lab incorporations but does confirm that lab incorporations in general are not dependent on experimental stimulations. Further studies could address how different stimulations during sleep, especially those leading to cortical arousal, may affect LIDs.”

--Comment 5

Line 196. If my understanding is correct, two of the raters are students (J2) and supervisor (J3). In this respect, even if they rated the reports “independently,” they may not constitute fully "independent” raters. This may explain why the inter-rater agreement was very high between these raters for most categories. Please include in table or text the level of inter-rater agreement with respect to the third rater, J4, and comment on relevant differences (if present) between J2/J3 scores and J4 scores.

--Reply 5

In fact both raters J2 and J4 were, or had been, students of J3. The scoring criteria were decided from a common agreement between the three raters before they each separately rated the dreams. We changed the term ‘independently’ for ‘separately’ to avoid any confusion (lines 202 and 208). 

We also added the ranges of inter-rater agreement estimates between J2/J4 and J3/J4 (Lines 213-216, additions in bold):

“Reliability estimates between each pair of judges were then calculated. Kappa estimates ranged between 0.189 – 0.873 (Mean = 0.646) for J2 and J3; between 0.364 – 0.843 (Mean = 0.577) for J2 and J4, and between 0.289 – 0.824 (Mean = 0.497) for J3 and J4. Scoring reliabilities between J2 and J3 were overall strongest.”

Large differences between the raters were mostly due to category definitions that were too vague. Those definitions were clarified between the judges when needed when working to consensus. We now clarify this process (line 222, additions in bold):

“For categories with low reliability estimates (Sleep activities: 0.468; Sensory: 0.454; Temporal orientation: 0.189, Observer Effect: 0.337), the 3 judges together refined the definition of those categories and worked through each dream report to a consensus; the consensus scores were used as dependent measures.”

--Comment 6

Lines 432-434. The authors may consider as explanation also potential differences in memory encoding between night and day experiments related to the level of arousal during the pre-sleep phase (e.g., Baddeley et al., 2007, Memory and Time of Day), rather than during actual sleep as they suggest (Lines 445-449).

--Reply 6

Thank you for the suggestion, we now consider this possible explanation in the Discussion (lines 450-453, addition in bold):

“This might especially be the case for morning naps, during which the repertoire of ‘previous day’ elements was limited in large part only to travel to the lab and the lab experience itself. Moreover, memory encoding has been found to be more efficient in the morning than later at night, possibly due to circadian fluctuation in levels of arousal (Baddeley et al. 2007), which could in turn increase the incorporation rates of pre-sleep events in morning nap dreams.”

--Comment 7

The Discussion section is extremely long and almost constitutes a distinct article. I would suggest to the authors to move parts of the Discussion - particularly the very long section regarding the themes of incorporation and their interpretation - in supplementary material and only include in the manuscript a shortened summary of the most relevant aspects. This would improve the overall readability of the manuscript.

--Reply 7

We made major cuts throughout the Discussion, especially in the section where the organizational themes are discussed: in total, we cut 28% (1486 out of 5230 words) from the original Discussion. We also took this opportunity to reorganize the sections on Methodological advantages and limitations (last section of the Discussion) for greater clarity. See the revised manuscript for details.

Reviewer #2 

--General comment

This study considered 528 dreams of 343 participants collected in a Montreal sleep lab aiming to investigate laboratory incorporations using a new scoring system. As main quantitative findings lab incorporations occurred in over a third of analyzed dreams and were mainly associated to REM sleep and morning naps. Concerning qualitative aspects, the study shows that themes associated with lab incorporation were: Meta-dreaming (e.g., lucid dreams and false awakenings, Sensory incorporations (27%), Wayfinding, Sleep as performance, Friends/Family in the lab and Being an object of observation. A relatively high proportion of dreams contained more dreams incorporation of projections into a near future than into the past. On the whole, I have a positive opinion on this study which provides relevant pieces of knowledge for studying dream recall in laboratory settings (i.e, a methodological merit), but also for increasing knowledge on the sources of dream per se. I have only minor points:

--Comment 1

Sample size is strength and weakness of this study, since many sources of variability have been included for collecting such sample: I am not completely convinced that the statistical design was successful in controlling all the sources of variability

--Reply 1

We agree that large datasets can have the disadvantage of introducing variability, but we believe that this is outweighed by the advantage of greater statistical power. While we did our best to control for inter-study sources of variability, controlling for each of nine projects separately (while also controlling for sleep stages and sleep timing) led to convergence problems in our statistical model. We now further discuss this limitation in the Discussion (see reply to Comment 4, Reviewer 1).

--Comment 2

Percentages of recall from NREM (64.3%) were quite lower compared to other studies: On the other hand, percentages upon awakenings from N3 seem unusually high (64.3%). In my opinion, the authors should make an effort to explain this finding and to discuss it in relation to previous studies.

--Reply 2

As reviewed in Nielsen (2000), the average dream recall from 25 studies using a similar liberal definition of dreams as we are (that includes any cognitive activity) estimates that 50.9% of NREM awakenings results in a mentation recall. Our result of 69% recall from NREM awakenings is in fact higher compared to those studies, but falls within the range of these previous studies (23-75%).

Percentage of dream recall from stage 3 or 4 were previously found to be quite high as well: “On average, recall from these stages is equal to that of stage 2 sleep; a tally of eight studies (Cavallero et al. 1992; Fein et al. 1985; Foulkes 1966; Lloyd & Cartwright 1995; Moffitt et al. 1982; Pivik 1971; Pivik & Foulkes 1968; Rotenberg 1993b) revealed an average recall rate of 52.5 ±18.6%” (Nielsen, 2000). But slightly higher dream recall from our N3 awakenings may be partly due to low sample sizes (only 14 awakenings in total) or to awakenings from morning naps which may increase dream recall percentages – however our recall rates for N3 also fall within the range of previous studies.

In order to meet the editor’s requirement that we shorten the Discussion (see Reviewer 1, Comment 7), we decided not to discuss dream recall rates as we consider that they fall within the normal range and are not the main focus of this study.

Nonetheless, we now clarify how dream reports were obtained by adding the following in Methods (lines 154-158):

“Dream reports were collected immediately following each awakening, either by recording oral reports (which were later transcribed) or by having participants directly type the reports on a keyboard in bed. A liberal definition of dream activity (anything that was going through your mind before waking up) was used in all studies.”

--Comment 3

Concerning the main finding related to the effect of sleep stage and morning naps, I am not completely convinced that performing Sleep stage or Sleep timing as the only predictor is sufficient to exclude an interaction between these basic factors. This issue should be discussed.

--Reply 3 

We could not directly compute an interaction between sleep stage and sleep timing as this led to a convergence problem in our GLM models. Our analyses showing that lab incorporations are more frequent in REM awakenings in both naps and overnight studies separately; and that lab incorporations are more frequent in naps awakenings from both REM and NREM awakenings separately suggest somewhat independent factors. However, we agree that it doesn’t fully exclude the possibility of some interaction. 

We now address this in the Results (lines 265-266 and 273-274, addition in bold):

“An interaction effect could not be directly computed because of a convergence problem introduced by the complexity of the model. However, when using Sleep stage as the only predictor, REM dreams were still more often LIDs than were NREM dreams in both nap …”

“These results suggest that stage and timing predictors influence LIDs independently (Fig 1B) even though an interaction between these factors is not fully excluded from these results.”

--Comment 4

The analyses also are not completely convincing in excluding that experimentally induced sleep arousal alone does not cause LIDs. Cortical arousal is a main determinant of the probability to have dream recall, and (different) experimental manipulations may have affected LIDs.

--Reply 4

We removed our claim that LIDs are not dependent on experimentally-induced sleep arousal (lines 488-494), and instead further discuss limitations of our sleep manipulation category in the Discussion (lines 780-787; please see reply to Comment #4, Reviewer 1).

---

## [Decision Letter · Decision Letter 1]

9 Sep 2021

Dreaming of the sleep lab

PONE-D-21-23305R1

Dear Dr. Nielsen,

We’re pleased to inform you that your manuscript has been judged scientifically suitable for publication and will be formally accepted for publication once it meets all outstanding technical requirements.

Kind regards,

Serena Scarpelli

Academic Editor

PLOS ONE

Additional Editor Comments (optional):

Reviewers' comments:

Reviewer's Responses to Questions

**Comments to the Author**

1. If the authors have adequately addressed your comments raised in a previous round of review and you feel that this manuscript is now acceptable for publication, you may indicate that here to bypass the “Comments to the Author” section, enter your conflict of interest statement in the “Confidential to Editor” section, and submit your "Accept" recommendation.

Reviewer #1: All comments have been addressed

Reviewer #2: All comments have been addressed

2. Is the manuscript technically sound, and do the data support the conclusions?

Reviewer #1: Yes

Reviewer #2: Yes

3. Has the statistical analysis been performed appropriately and rigorously? 

Reviewer #1: Yes

Reviewer #2: Yes

4. Have the authors made all data underlying the findings in their manuscript fully available?

Reviewer #1: No

Reviewer #2: No

5. Is the manuscript presented in an intelligible fashion and written in standard English?

Reviewer #1: Yes

Reviewer #2: Yes

6. Review Comments to the Author

Reviewer #1: The authors adequately addressed the raised concerns and significantly improved their manuscript through the revision process.

Reviewer #2: (No Response)

7. PLOS authors have the option to publish the peer review history of their article (what does this mean?). If published, this will include your full peer review and any attached files.

Reviewer #1: No

Reviewer #2: **Yes: **Luigi De Gennaro

---

## [Editor Report · Acceptance letter]

21 Sep 2021

PONE-D-21-23305R1 

Dreaming of the sleep lab 

Dear Dr. Nielsen:

I'm pleased to inform you that your manuscript has been deemed suitable for publication in PLOS ONE. Congratulations! Your manuscript is now with our production department. 

Kind regards, 

on behalf of

Dr. Serena Scarpelli 

Academic Editor

PLOS ONE